# Hepcidin Upregulation in Colorectal Cancer Associates with Accumulation of Regulatory Macrophages and Epithelial–Mesenchymal Transition and Correlates with Progression of the Disease

**DOI:** 10.3390/cancers14215294

**Published:** 2022-10-27

**Authors:** Antonio Di Grazia, Davide Di Fusco, Eleonora Franzè, Marco Colella, Georgios Strimpakos, Silvia Salvatori, Vincenzo Formica, Federica Laudisi, Claudia Maresca, Alfredo Colantoni, Angela Ortenzi, Carmine Stolfi, Ivan Monteleone, Giovanni Monteleone

**Affiliations:** 1Department of Systems Medicine, University of Rome “Tor Vergata”, 00133 Rome, Italy; 2Institute of Biochemistry and Cell Biology (IBBC), National Council of Research (CNR), 00146 Rome, Italy; 3Gastroenterology Unit, Fondazione Policlinico “Tor Vergata”, 00133 Rome, Italy; 4Medical Oncology Unit, Fondazione Policlinico “Tor Vergata”, 00133 Rome, Italy; 5Department of Biomedicine and Prevention, University of Rome “Tor Vergata”, 00133 Rome, Italy

**Keywords:** type 2 macrophages, cancer metastasis, prognosis, Stat3, cytokines

## Abstract

**Simple Summary:**

Colorectal cancer (CRC) is one of the deadliest cancers worldwide, and this is mainly due to the fact that advanced, metastatic CRC is poorly responsive to current pharmacologic treatment. Therefore, studies aimed at dissecting the mechanisms driving CRC progression are worth pursuing in order to improve the treatment of the metastatic disease. We here analyzed whether hepcidin, a peptide hormone involved in the growth of cancer cells in many organs, identifies some subsets of CRC with advanced and aggressive course. By integrating results of in vitro/ex vivo studies with data of bioinformatics databases, we show that CRC cells express high levels of hepcidin and provide data supporting the role of hepcidin in promoting a tumor microenvironment that sustains CRC cell growth and metastasis. Overall, these novel findings could help develop therapeutic strategies to combat metastatic CRC.

**Abstract:**

Advanced, metastatic colorectal cancer (CRC) is associated with high rate of mortality because of its poor responsiveness to chemotherapy/immunotherapy. Recent studies have shown that hepcidin, a peptide hormone produced mainly by hepatocytes, is expressed by and enhances the growth of tumor cells. We here assessed whether hepcidin expression helps identify subsets of CRC with advanced and aggressive course. By integrating results of in vitro/ex vivo studies with data of bioinformatics databases, we initially showed that hepcidin RNA and protein expression was more pronounced in tissue samples taken from the tumor area, as compared to the macroscopically unaffected, adjacent, colonic mucosa of CRC patients. The induction of hepcidin in the colonic epithelial cell line HCEC-1ct by interleukin (IL)-6, IL-21 and IL-23 occurred via a Stat3-dependent mechanism and, in primary CRC cells, hepcidin co-localized with active Stat3. In CRC tissue, hepcidin content correlated mainly with macrophage accumulation and IL-10 and CD206 expression, two markers of regulatory macrophages. Consistently, both IL-10 and CD206 were up-regulated by hepcidin in blood mononuclear cells. The highest levels of hepcidin were found in metastatic CRC and survival analysis showed that high expression of hepcidin associated with poor prognosis. Moreover, hepcidin expression correlated with markers of epithelial-to-mesenchymal transition and the silencing of hepcidin in CRC cells reduced epithelial-to-mesenchymal transition markers. These findings indicate that hepcidin is markedly induced in the advanced stages of CRC and suggest that it could serve as a prognostic biomarker in CRC.

## 1. Introduction

Colorectal cancer (CRC), one of the most frequent neoplasms worldwide, is a multifactorial disease, which develops in the majority of cases sporadically because of acquired somatic mutations and aberrant epigenetic alterations in genes controlling cell growth and survival [1,2]. Several environmental and lifestyle habits have been associated with CRC development and epidemiological studies showed that various dietary factors can either increase or reduce the risk of cancer [3,4,5,6]. Although screening programs help identify neoplastic lesions, CRC remains one of the most deadly cancers worldwide mainly due to the fact that advanced CRC is poorly responsive to current pharmacologic treatment [7,8]. This highlights the need of further work to better dissect the mechanisms driving CRC initiation and progression in order to both improve primary prevention and treat the metastatic disease.

Hepcidin is a peptide hormone produced mainly in the liver by hepatocytes but also in other organs and tissues by different cell types (e.g., macrophages and enterocytes) [9,10]. Although hepcidin was initially considered as an anti-microbial factor, it acts mainly as a regulator of iron homeostasis through the hepcidin–ferroportin (FPN1) axis [11,12]. The peptide is synthesized as pre-prohepcidin, which is cleaved to produce prohepcidin, comprising 60 amino acids. Prohepcidin is then cleaved by furin-like prohormone convertases to generate the two mature isoforms of 20 and 25 amino acids, respectively. The carboxy terminus of the mature hepcidin, which comprises four highly conserved disulfide bonds, forms a β-hairpin, while the N-terminus is essential for FPN1 interaction. The binding of hepcidin to membrane-bound FPN1 induces FPN1 internalization and degradation, thus, reducing iron availability to tissues [13,14,15].

The dysregulation of the hepcidin–FPN1 axis has been associated with the development of various cancers [16]. For instance, hepcidin deficiency increases the susceptibility for developing liver fibrosis, which is a risk factor for hepatocellular carcinoma (HCC) and reduced hepatic hepcidin expression occurs in patients with cirrhosis, which is also a risk factor for HCC development [17]. Moreover, hepcidin promoter DNA is hyper-methylated in HCC, resulting in its transcriptional repression [18]. On the other hand, reduced hepatic hepcidin expression was found to be protective against the progression of lung and breast cancer [19]. Moreover, in patients with breast cancer, hepcidin expression is significantly upregulated in tissues and serum and there is evidence supporting the role of hepcidin in the development of the malignant phenotype and resistance to doxorubicin [20,21,22]. In lung cancer, hepcidin acts as an oncogene by promoting the development and progression, while in pancreatic cancer and prostate cancer hepcidin expression has been associated with vascular invasion [23,24,25]. A more pronounced expression of hepcidin RNA was also found in CRC samples, as compared to matched normal colonic samples, and treatment of the CRC cell line HT-29 with hepcidin induced FPN1 degradation, thus, promoting iron sequestration inside the cells with the downstream effect of increasing cell survival [26]. Recently, by using a sporadic model of CRC, Schwartz and colleagues showed that mice that are deficient in hepcidin, specifically in the colon tumor epithelium, exhibited diminished tumor number, burden and size as a result of the ability of hepcidin to degrade FPN1 and sequester iron, thus, sustaining the production of nucleotides and CRC cell proliferation [27]. Altogether these observations raise the possibility that hepcidin may play a role in CRC, even though little is known about the involvement of hepcidin in the different phases of colon tumorigenesis.

In the present work, we assessed whether hepcidin expression marks subsets of CRC with advanced and more aggressive course.

## 2. Materials and Methods

### 2.1. Patients and Samples

Paired tissue samples were taken from the tumoral area and the macroscopically unaffected, adjacent, colonic mucosa of 26 patients who underwent colon resection for sporadic CRC at the Tor Vergata University Hospital (Rome, Italy). Patients with sporadic CRC received neither radiotherapy nor chemotherapy before surgery. Written informed consent was obtained from all the patients. The study protocol was approved by the Comitato Etico Indipendente, Fondazione PTV, Policlinico Tor Vergata, Rome, Italy (protocol number 129/17).

### 2.2. Cell Culture

All reagents were from Sigma-Aldrich (Milan, Italy), unless specified. The human CRC cell lines HCT-116, HT-29 and DLD-1 were obtained from the American Type Culture Collection (ATCC, Manassas, VA, USA). HCT-116 (with microsatellite instability, BRAF, PTEN and TP53 wild type, and KRAS G13D and PIK3CA H1047R mutations) and HT-29 (with PTEN and KRAS wild type, and TP53 R273H, BRAF V600E and PIK3CA P449T mutations) were maintained in McCoy’s 5A medium supplemented with 10% fetal bovine serum (FBS) and 1% penicillin/streptomycin (P/S) (Lonza, Verviers, Belgium) while DLD-1 (with microsatellite instability, BRAF and PTEN wild type; TP53 S241F, KRAS G13D and PIK3CA E545K; and D549N mutations) was maintained in RPMI 1640 medium supplemented with 10% FBS and 1% P/S. The normal human colon epithelial cell line HCEC-1ct was obtained from EVERCYTE GmbH (Vienna, Austria) and cultured in ColoUp medium (EVERCYTE GmbH). Cell lines were authenticated by STR DNA fingerprinting using the PowerPlex 18D System kit, according to the manufacturer’s instructions (Promega, Milan, Italy). The STR profiles of all the cell lines matched the known DNA fingerprints. To investigate whether interleukin (IL)-6, IL-21 and IL-23 modulate hepcidin expression, HCEC-1ct cells (2.5 × 10^5^) were stimulated with recombinant human IL-6 (Peprotech, London, UK), IL-21 and IL-23 (R&D Systems, Minneapolis, MN, USA) (all used at 20 ng/mL). After 20 h, hepcidin expression was assessed by Western blotting. To determine the contribution of Stat3 on cytokine-induced hepcidin expression, HCEC-1ct (1 × 10^5^) cells were transfected with either a Stat3 antisense oligonucleotide (AS) (5′-AGCTGATTCCATTGGGCCAT-3′) or control oligonucleotide with sense orientation (S) (both used at a final concentration of 200 nmol/L) for 24 h using Opti-MEM medium and Lipofectamine 3000 reagent (both from Life Technologies, Milan, Italy) and then stimulated with IL-6, IL-21 and IL-23 for further 20 h. To determine whether hepcidin controls expression of mesenchymal markers, HCT-116 (2.5 × 10^5^) cells were transfected with either hepcidin siRNA (final concentration 10–25 nmol/L; Santa Cruz Biotechnology, Santa Cruz, CA, USA) or control siRNA (final concentration 10 nmol/L; Santa Cruz Biotechnology) for 24 h using Opti-MEM medium and Lipofectamine 3000 reagent. In further experiments, HCT116 cells were either left untreated or stimulated with exogenous hepcidin (1000 and 2000 ng/mL, Creative Biomart, Shirlay, NY, USA) for 30 min or 24 h. Peripheral blood mononuclear cells (PBMCs) were isolated from 6 healthy volunteers through density gradient centrifugation (Lympholyte-H; Cedarlane Labs, Burlington, ON, Canada) and cultured (1 × 10^6^) in RPMI 1640 medium supp1lemented with 10% FBS and 1% P/S in the absence or presence of exogenous hepcidin (final concentration 100–200 ng/mL) or interleukin-13 (IL-13) (final concentration 50 ng/mL; Creative Biomart, Shirlay). After 24 and 48 h, cells were used to analyze IL-10, CD163, CD206 and TNF by real-time (RT)-PCR and flow-cytometry, respectively.

### 2.3. Protein Extraction and Western Blotting

Total proteins were isolated in lysis buffer containing 0.1 mmol/L EDTA, 10 mmol/L HEPES (pH 7.9), 10 mmol/L KCl, 0.5% Nonidet P40, 0.2 mmol/L ethylene glycol-bis (β-aminoethyl ether)-N,N,N’,N’-tetraacetic acid and complemented with 10 mg/mL leupeptin, 1 mmol/L dithiothreitol, 10 mg/mL aprotinin, 1 mmol/L phenylmethylsulfonyl fluoride, 1 mmol/L NaF and 1 mmol/L Na3VO4. Lysates were clarified by centrifugation and analyzed by sodium dodecyl sulfate polyacrylamide gel electrophoresis (SDS-PAGE). After being transferred on nitrocellulose membranes, they were incubated with antibodies against anti-human hepcidin (final dilution 1:1000, Abcam, Cambridge, UK), p-Stat3 (final dilution 1:1000, Cell Signaling, Danvers, MA, USA) and Stat3 (final dilution 1:1000, Santa Cruz Biotechnology), followed by a secondary antibody conjugated to HRP (Dako, Agilent Technologies, Santa Clara, CA, USA). A mouse anti-β-actin antibody was used to detect β-actin. Densitometry scanning was used to analyze the intensity of the immunoreactive bands.

### 2.4. Immunohistochemistry and Immunofluorescence

Immunohistochemistry was performed on formalin-fixed, paraffin-embedded sections of normal and tumoral samples of CRC patients. Sections were deparaffinized and dehydrated through xylene and ethanol, and the antigen retrieval was performed in Tris-EDTA citrate buffer (pH 7.8) for 30 min in a thermostatic bath at 98 °C (Dako Agilent Technologies, Santa Clara, CA, USA) Immunohistochemical staining was performed using a monoclonal antibody directed against human hepcidin (final dilution 1:500; SAB) incubated at room temperature for 1 h, followed by a biotin-free horse radish peroxidase (HRP) polymer detection technology with 3,30 diaminobenzidine as a chromogen MACH 4 Universal HRP-Polymer Kit (Biocare Medical, Pacheco, CA, USA).

Frozen sections of mucosal samples were embedded in a cryostat mounting medium (Neg–50 Frozen Section Medium, Thermo Fisher Scientific, Waltham, MA, USA), snap-frozen, and stored at −80 °C. Sections (6-µm thick) were mounted onto superfrost glass slides. Frozen sections of mucosal samples and in HCT-116 cells were fixed by 3.7% formaldehyde for 10 min at 4 °C and permeabilized with 0.1% Triton for 10 min at room temperature. To prevent non-specific labeling, sections were incubated with bovine serum albumin 1%, Tween 0.1%, and glycine 2% for 1 h at room temperature and then with anti-hepcidin (final dilution 1:100; Abcam, Cambridge, UK), anti-pStat3 (final dilution 1:100; Santa Cruz Biotechnology), anti-α-smooth muscle actin (α-SMA) (final dilution 1:100; Thermo Fisher Scientific) and anti-CD31 (final dilution 1:100; Thermo Fisher Scientific). After washing with phosphate-buffered saline (PBS1X) 3 times, the sections were incubated with a goat anti-rabbit Alexa 488 (final dilution 1:2000; Invitrogen, Carlsbad, CA, USA) or goat anti-mouse Alexa 568 (final dilution 1:2000; Invitrogen) secondary antibody for 1 h at room temperature. Cells were washed with PBS1X 3 times and mounted using Prolong gold antifade reagent with DAPI (Invitrogen). Confocal laser scanning microscopy (Olympus confocal IX83) was used to assess positive cells.

Moreover, HCT116 cells cultured as above were incubated with monoclonal antibodies against hepcidin (final dilution 1:100), α-SMA (final dilution 1:100), CD31 (final dilution 1:100), Vimentin (final dilution 1:100; Thermo Fisher Scientific), Collagen I (COL1A1) (final dilution 1:100; Novus, Milan, Italy) and Collagen III (COL3A1) (final dilution 1:100; Novus, Milan, Italy) overnight at 4 °C and then with the above secondary antibodies. Finally, cells were washed with PBS1X 3 times, mounted using Prolong gold antifade reagent with DAPI (Invitrogen) and visualized using a Leica DMI4000 B microscope with Leica application suite software (V4.6.2) (Wetzlar, Germany).

### 2.5. RNA Extraction and RT-PCR

RNA was extracted using PureLink mRNA mini kit (Thermo Fisher Scientific, Waltham, MA, USA), according to the manufacturer’s instructions. RNA (1 µg per sample) was reverse transcribed into complementary DNA (cDNA) by M-MLV Reverse Transcriptase and this was amplified using the following conditions: denaturation for 1′ minute at 95 °C; annealing for 30 s at 60 °C for β-actin, hepcidin, α-SMA, COL1A1, and COL3A1; 62 °C for IL-10; 58 °C for Vimentin; 30 s of extension at 72 °C. TNF, CD206 and CD163 were evaluated using commercial TaqMan probes (Applied Biosystems, Foster City, CA, USA). RNA expression was calculated relative to the housekeeping β-actin gene on the base of the ΔΔCt algorithm. Sequences of the primers used were as follows: β-actin: Forward: 5′- AAGATGACCCAGATCATGTTTGAGACC-3′; Reverse: 5′-AGCCAGTCCAGACGCAGGAT-3′; hepcidin: Forward: 5′-CACAACAGACGGGACAACTT-3′; Reverse: 5′-CGCAGCAGAAAATGCAGATG-3′; α-SMA: Forward: 5′-TCTGGAGATGGTGTCACCCA 3′; Reverse: 5′-AATAGCCACGCTCAGTCAGG-3′; COL1A1: Forward: 5′-GGACACAGAGGTTTCAGTGG-3′; Reverse: 5′-CCAGTAGCACCATCATTTCC-3′; COL3A1; Forward: 5′-GGAGAATGTTGTGCAGTTTGC-3′; Reverse: 5′-TCTGAGGACCAGTAGGGCAT-3′; IL-10: Forward: 5′-GGC ACCCAGTCTGAGAACAG-3′; Reverse: 5′-CTTGG CAACCCAGGTAACCC-3′; Vimentin: Forward: 5′ -GGAGCAGCAGAATAAGATCC-3′; Reverse: 5′-CCAGAGACGCATTGTCAAC-3′.

### 2.6. Flow Cytometry

PBMCs were stained with the monoclonal human antibodies against IL-10 (Alexa Fluor 488, PE, eBioscience, San Diego, CA, USA) or CD206 (Alexa Fluor 647, APC, Becton Dickinson, Milan, Italy). Both antibodies were used at 1:50 final dilution. To assess IL-10-producing cells, phorbol myristate acetate (final concentration 40 ng/mL), Ionomycin (final concentration 1 mg/mL) and Brefeldin A (final concentration 10 mg/mL) were added 4 h before the end of the culture. In all experiments, appropriate isotype control IgG was used. Analysis was performed using the Kaluza software (Beckman Coulter Life Sciences, Pasadena, CA, USA).

### 2.7. UALCAN

UALCAN (http://ualcan.path.uab.edu/ (accessed on 1 May 2022)), a web-based tool that provides in-depth analyses of cancer OMICS data (TCGA, MET500, CPTAC and CBTTC), was used to assess RNA hepcidin expression in normal and CRC samples and the association between hepcidin and the following parameters: sex, race, age, body weight, cancer stages, nodal metastasis status of patients with CRC.

### 2.8. Gene Expression Profiling Interactive Analysis (GEPIA)

GEPIA (http://gepia.cancer-pku.cn/index.html (accessed on 1 May 2022)), a web portal for gene expression analysis based on TCGA and GTEx data, was used to investigate the relationship between hepcidin expression and macrophages markers, epithelial-to-mesenchymal transition markers or survival analysis in CRC patients.

### 2.9. Tumor Immune Estimation Resource (TIMER)

TIMER (https://cistrome.shinyapps.io/timer/ (accessed on 1 May 2022)) is an interactive web portal that allows one to analyze the tumor infiltration levels of different immune cells. In the present study, we analyzed the correlation between hepcidin expression and the infiltration level of CRC with macrophages, B cells, CD8+ T cells and CD4+ T cells.

### 2.10. Statistical Analysis

Differences between groups were compared using the Student’s *t*-test or Mann–Whitney test. Correlation between hepcidin expression and immune cell types, macrophage and epithelial-to-mesenchymal transition markers was evaluated using the Spearman correlation. ANOVA was used to compare hepcidin levels in the different CRC stages. Correlation between hepcidin expression and overall survival and free-disease survival was assessed with the Kaplan–Meier curve. Differences were considered to be statistically significant for a *p* value < 0.05. All analyses were made using GraphPad Prism version 5.00 software for Windows (GraphPad Software, San Diego, CA, USA [www.graphpad.com (accessed on 1 September 2021–1 October 2022)]).

## 3. Results

### 3.1. Hepcidin Is Up-Regulated in Human Sporadic CRC

By Western blotting, we initially assessed hepcidin expression in paired tissue samples taken from the tumor area and the adjacent, normal, colonic mucosa of patients with sporadic CRC. A more pronounced expression of hepcidin was seen in the tumor areas and densitometry analysis of Western blots showed a significant up-regulation of hepcidin in the tumoral samples, as compared to the normal samples (Figure 1a and Appendix A). These findings were confirmed by immunohistochemistry which showed that the percentages of hepcidin-positive cells were significantly higher in the tumor area, as compared with the non-tumoral area of the same patients (Figure 1b). Moreover, RT-PCR analysis showed that hepcidin RNA transcripts were significantly higher in tumoral samples, compared with normal samples (Figure 1c). Analysis of hepcidin-expressing cells in the non-tumoral areas by confocal microscopy showed that both α-SMA- and CD31-positive cells expressed hepcidin (Appendix A). Hepcidin protein expression was significantly upregulated in CRC cell lines (i.e., DLD-1, HCT-116 and HT-29), as compared to that in HCEC-1ct (Figure 1d; Appendix A).

To support these findings, we assessed hepcidin RNA expression in 286 human CRC samples using the online database UALCAN. Relative to 41 cases of normal tissues, hepcidin RNA transcripts were significantly upregulated in CRC samples (Figure 2a). Moreover, high hepcidin RNA expression was seen in CRC samples from both males and females independently of the ethnicity of the patients, as compared with the corresponding normal samples (Figure 2b,c). Similarly, elevated hepcidin RNA was seen in CRC patients from different age groups independently of their body weight, as compared to the corresponding control groups (Figure 2d,e). CRC patients with extreme obesity had greater hepcidin RNA content than controls, but the difference was not statistically significant, probably reflecting the small number of CRC patients with extreme obesity included in the analysis (Figure 2e).

### 3.2. Stat3 Regulates Positively Hepcidin Production

Previous studies conducted mainly in human liver cells showed that IL-6, a cytokine over-produced in CRC, activates the transcription factor Stat3, thereby enhancing hepcidin expression [12,28]. To assess whether, in the human colon, activation of Stat3 increases hepcidin synthesis, HCEC-1ct cells were stimulated with the Stat3-activating cytokines IL-6, IL-21 and IL-23. After 20 h, total proteins were extracted and analyzed for hepcidin content by Western blotting. All these cytokines significantly increased hepcidin protein expression (Figure 3a). To confirm that the cytokine-driven induction of hepcidin occurred via a Stat3-dependent mechanism, HCEC-1ct cells were transfected with a specific Stat3 antisense (AS) or sense (S) oligonucleotide and after 24 h stimulated with the above cytokines for further 20 h. The knockdown of Stat3 prevented the cytokine-mediated hepcidin induction (Figure 3b). Consistently, in human CRC specimens, there was a co-localization of pStat3 (Tyr 705) and hepcidin in cancer cells (Figure 3c). To examine whether in CRC cells there is an autocrine regulation of hepcidin expression, HCT116 cells were treated with exogenous hepcidin and then evaluated for hepcidin content. The stimulation of HCT-116 cells with hepcidin-enhanced Stat3 phosphorylation and hepcidin expression is shown in Appendix A.

### 3.3. Hepcidin Promotes Induction of Regulatory Macrophage-Associated Markers

Next, by using TIMER online tool, we analyzed the correlation between hepcidin expression and infiltrating immune cells in CRC tissue. Hepcidin RNA expression correlated mainly with macrophage infiltration and to a lesser extent with both T cells and B cells (Figure 4a). The most abundant immune cells in solid tumors are macrophages [29]. In neoplastic progression two types of macrophage: tumor-killing and tumor-promoting macrophage subpopulations can be present [30,31]. By using the GEIPIA online tool, we investigated the interrelationship between hepcidin expression and markers of inflammatory and regulatory macrophages. There was a positive correlation between hepcidin RNA expression and markers associated with either inflammatory (i.e., TNF and IL-23A) or regulatory macrophages (i.e., IL-10 and CD206), even though the stronger correlation was seen with IL-10 and CD206 (Figure 4b,c).

Based upon these observations, we explored the possibility that hepcidin could influence the activation of regulatory macrophages. To this end, PBMCs were stimulated with exogenous hepcidin or IL-13, an inducer of regulatory macrophages, and analyzed for the content of IL-10, CD206 and CD163. As expected IL-13 enhanced differentiation of regulatory macrophages (Figure 5a). Similarly, the treatment of PBMC with hepcidin enhanced IL-10 and CD206 RNA expression (Figure 5a). Moreover, hepcidin treatment increased the percentage of both IL-10- and CD206-expressing PBMC (Figure 5b,c). In line with this, hepcidin also enhanced CD163 RNA expression (Appendix A).

### 3.4. High Hepcidin Expression Associates with Low Survival of CRC Patients

Next, by using the UALCAN online tool, we investigated whether hepcidin expression correlates with the different CRC stages, lymph node metastases and overall survival. There was no significant difference in terms of hepcidin RNA content between stage 1 of CRC and normal controls. In contrast, patients with stages 2, 3 and 4 exhibited levels of hepcidin RNA greater than that measured in controls (Figure 6a). Significantly higher hepcidin RNA levels were seen in CRC patients, compared to controls independently of the lymph node involvement (Figure 6b). The Kaplan–Meier curve from GEIPIA database showed that patients with higher hepcidin expression had a significantly reduced overall survival and disease-free survival, compared with patients with low hepcidin levels (Figure 6c–f).

Epithelial-to-mesenchymal transition is a process that plays a roles in promoting carcinoma invasion and metastasis [32,33,34,35]. So, we investigated the relationship between hepcidin expression and epithelial-to-mesenchymal transition markers by using GEIPIA tool. A negative or very poor correlation was observed between hepcidin RNA expression and several epithelial markers (Figure 7a). In contrast, there was a positive correlation between hepcidin RNA expression and mesenchymal markers (Figure 7b).

Consistently, the silencing of hepcidin in HCT-116 cells led to a significant reduction of the mesenchymal markers (Figure 8a). At the same time point, hepcidin silencing did not significantly alter the HCT-116 cell shape (Figure 8b).

## 4. Discussion

Despite advances in the strategies adopted to make early diagnosis, CRC remains a leading cause of cancer-related death in both males and females worldwide, and this is because CRC is often diagnosed at an advanced stage, which is poorly responsive to chemotherapy/immunotherapy [36]. Therefore, studies aimed at both clarifying the factors/mechanisms that promote CRC cell diffusion and metastasis and identify prognostic biomarkers are worth pursuing. The present study was undertaken to evaluate whether hepcidin expression marks distinct subsets of CRC. By real-time PCR and Western blotting, we initially documented the over-expression of hepcidin in CRC, as compared to control samples, and this was evident at both RNA and protein level. Immunohistochemistry and confocal microscopy confirmed up-regulation of hepcidin in CRC and showed that cancer cells were the major source of the protein in the tumor area. Moreover, a more pronounced expression of hepcidin was seen in three different CRC cell lines, as compared to the normal colonic epithelial cell line HCEC-1ct. These findings were confirmed by analysis of various bioinformatics databases, which showed enhanced RNA expression of hepcidin in CRC independently of the clinical characteristics of the patients. Altogether, these data confirm and expand on results of previous studies reporting up-regulation of hepcidin in CRC [37]. For example, Ward and colleagues showed that urinary hepcidin levels were positively associated with increasing T-stage of CRC and hepcidin RNA was expressed in 34% of CRC tissue specimens [37]. Along the same line are the results of Sornjai and colleagues, who showed that hepcidin RNA was detected in 38% (8 out of 21) of CRC tissue samples, as compared to 29% (6 out of 21) of normal matched tissue samples [26]. More recently, it was shown that the colon tumor epithelium-derived hepcidin establishes an axis to sequester iron in order to maintain the nucleotide pool and sustain proliferation of CRC cells [27].

The factors/mechanisms that drive hepcidin synthesis in CRC remain unknown. Since induction of hepcidin in other systems has been associated with activation of Stat3, we explored the possibility that this transcription factor could sustain hepcidin production in CRC cells [28]. Indeed, Stat3 is over-expressed in human CRC, where it is supposed to play a major role in the growth, survival and diffusion of malignant cells [38]. Activation of Stat3 with IL-6, IL-21 and IL-23, the production of which is up-regulated in CRC tissue [39,40,41], was accompanied by enhanced hepcidin induction in colonic epithelial cells, and the knockdown of Stat3 with a specific antisense oligonucleotide abrogated the cytokine-driven hepcidin synthesis. Moreover, in CRC tissue, there was a co-localization of Stat3 and hepcidin. These observations raise the possibility that different cytokines released by immune cells infiltrating the CRC tissue activate Stat3 in cancer cells, thereby promoting hepcidin synthesis. Our findings do not, however, exclude the possibility that induction of hepcidin in CRC cells may also rely on additional factors/mechanisms. Indeed, it is well known that the expression of hepcidin is regulated by iron excess and hypoxia [42]. Moreover, hepcidin can be upregulated in mouse primary hepatocytes from both wild-type and IL-6 knockout mice by IL-1 [43]. Our data also suggest that in human CRC there is an autocrine regulation of hepcidin, as stimulation of HCT116 cells with exogenous hepcidin increased the activation of Stat3 and expression of hepcidin.

The CRC microenvironment contains a mixture of immune cells, which together with stromal cells, can either restrict or enhance CRC cell growth, survival and diffusion, as well as influence the outcome of chemotherapy and immunotherapy [44,45,46]. Specifically, activation of regulatory macrophages induces secretion of proangiogenic and growth factors, as well as causing suppression of T cell effector function by releasing immunosuppressive cytokines and affecting their metabolism [30,47,48]. By using TIMER and GEIPIA tools, we documented in CRC samples a positive correlation between the expression of hepcidin and accumulation of macrophages and expression of IL-10 and CD206, two markers of regulatory macrophages. Although, these findings do not help to understand the role of hepcidin in the recruitment and differentiation of pro-tumorigenic macrophages, our in vitro data suggest that hepcidin can contribute to the activation of such cells. Indeed, stimulation of blood mononuclear cells with exogenous hepcidin led to enhanced IL-10 and CD206 RNA expression and increased percentages of IL-10- and CD206-expressing cells. These data are consistent with results of previous in vitro and in vivo studies, showing that hepcidin inhibits the activation of inflammatory, anti-tumorigenic macrophages and suppresses LPS-induced IL-6 and TNF [49]. It is, thus, conceivable that CRC cell-derived hepcidin can contribute to generate a microenvironment that suppresses antitumor immunity and promotes tumor progression. This hypothesis is supported by the demonstration that the greatest levels of hepcidin RNA were found in patients with metastatic CRC and CRC patients with high hepcidin expression exhibited a markedly worse survival rate than those with low expression of hepcidin. Our findings are in line with the demonstration that patients with pancreatic cancer and who express high levels of hepcidin have a worse overall survival than patients with low or absent hepcidin expression [25]. Moreover, hepcidin RNA expression was found to be higher in patients with metastatic renal cell carcinoma (RCC) than in those without metastasis, and high hepcidin expression correlated significantly with poor survival in RCC patients [50].

In CRC, there was a positive correlation between hepcidin expression and several markers of epithelial-to-mesenchymal transition, a phenomenon that plays a major role in cancer diffusion and metastasis [51,52]. Consistently, the silencing of hepcidin in CRC cells markedly reduced the expression of such markers.

## 5. Conclusions

Our study shows a preferential induction of hepcidin in the advanced stages of CRC. Further work is needed to assess whether hepcidin is a target for therapeutic intervention in patients with advanced CRC.

## Figures and Tables

**Figure 1 cancers-14-05294-f001:**
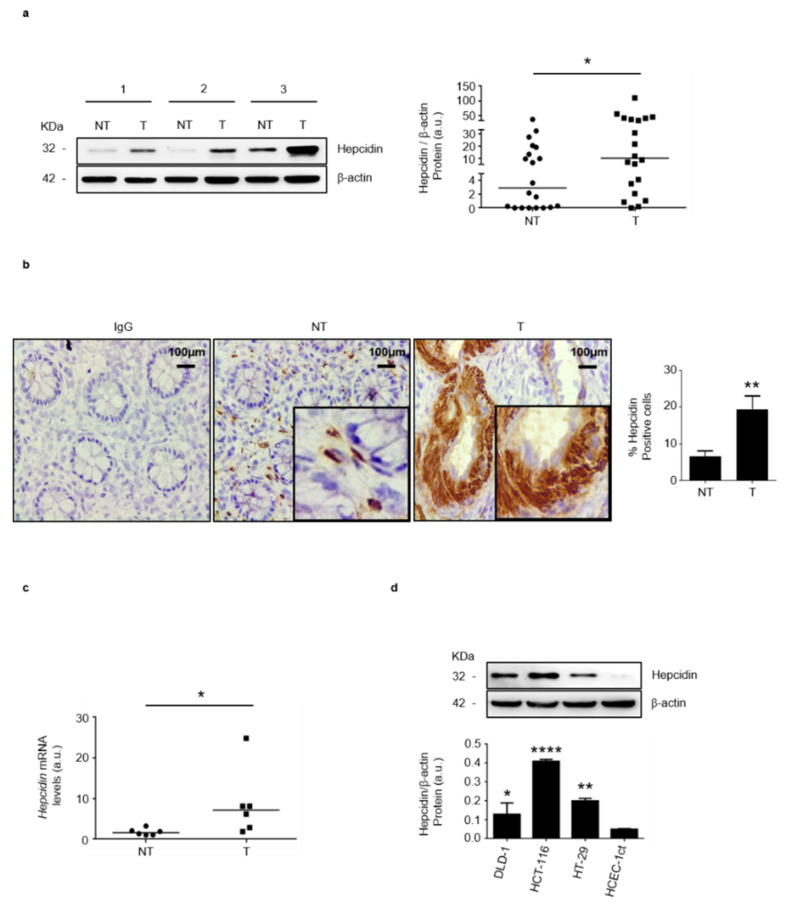
Hepcidin expression is increased in CRC. (**a**) Hepcidin protein expression was evaluated in paired tissue samples taken from the tumor area (T) and the adjacent, non-tumor area (NT) of CRC patients by Western blotting. β-actin was used as a loading control. Left panel: representative Western blots of hepcidin and β-actin in 3 different patients; right panel shows the quantitative analysis of hepcidin/β-actin protein ratio, as measured by densitometry scanning of Western blots. Each point in the graph indicates the value of hepcidin/β-actin in a single patient. Values are expressed in arbitrary units (a.u.); NT versus T, * *p*< 0.05; *n* = 20). (**b**) Representative images of immunohistochemistry of colon sections of paired tissue samples taken from the T and NT areas of CRC patients. Staining with a control isotype IgG is also shown. Scale bars, 100 μm. Inset shows staining at higher magnification. Right inset: the percentages (%) of hepcidin-positive cells in sections taken from the NT and T areas of 6 CRC patients. Data are expressed as mean ± SD (** *p*< 0.01). (**c**) Hepcidin RNA expression was evaluated in paired tissue samples taken from the T and NT areas of CRC patients by RT-PCR, and values were normalized to β-actin. Each point in the graph represents the value of hepcidin mRNA in a single patient (* *p* < 0.05; *n* = 6). (**d**) Hepcidin protein expression in CRC cell lines (DLD-1; HCT-116; HT-29) and HCEC-1ct cells evaluated by Western blotting. Representative Western blots and quantitative analysis of hepcidin/β-actin protein ratio, as measured by densitometry scanning of Western blots are shown. In the lower panel, values are expressed in arbitrary units (a.u.) and indicate mean ± SD of all experiments; DLD-1 versus HCEC-1ct, * *p* < 0.05; HCT-116 versus HCEC-1ct, **** *p*< 0.0001; HT-29 versus HCEC-1ct, ** *p* < 0.01; *n* = 3).

**Figure 2 cancers-14-05294-f002:**
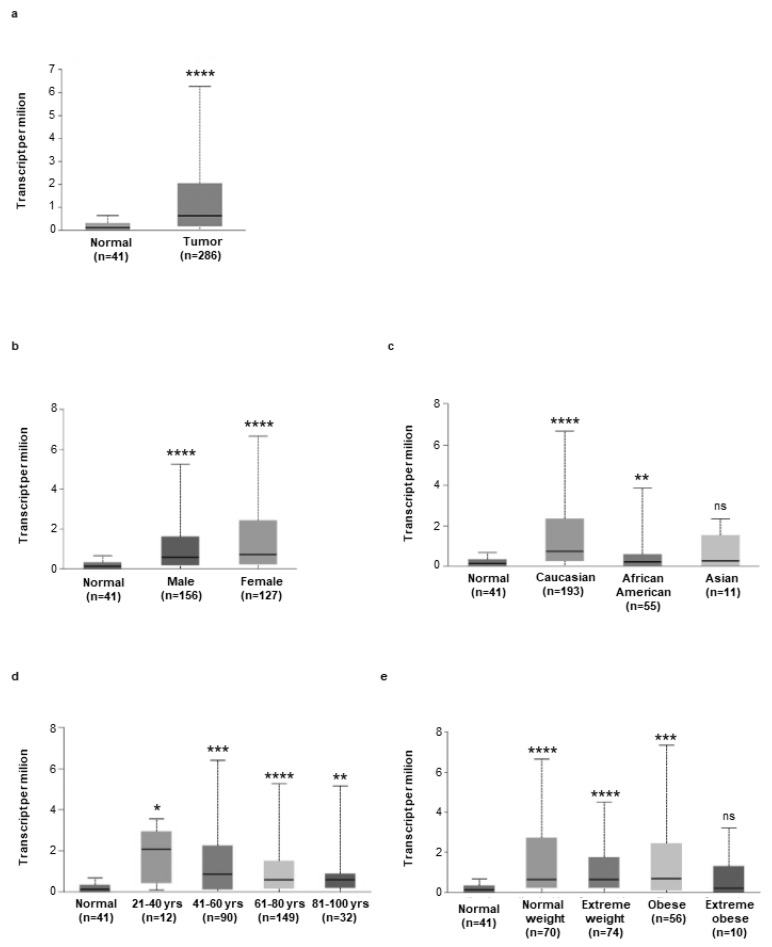
(**a**) Expression of hepcidin in colorectal cancer samples (*n* = 256) and normal tissue samples (*n* = 41), as evaluated by using the UALCAN database (cancer vs. normal (**** *p* < 0.0001). (**b**–**e**) Expression of hepcidin RNA in different groups of CRC based on patients’ gender (**b**) [Normal (*n* = 41) versus CRC male (*n* = 156), **** *p* < 0.0001; Normal (*n* = 41) versus CRC female (*n* = 127), **** *p* < 0.0001], patients’ ethnicity (**c**) [Normal (*n* = 41) versus CRC Caucasian (*n* = 193), **** *p* < 0.0001; Normal (*n* = 41) versus CRC African American (*n* = 55), ** *p* < 0.01; Normal (*n* = 41) versus CRC Asian (*n* = 10), ns]; patients’ age (**d**), [Normal (*n* = 41) versus CRC 21–40 years (*n* = 12), * *p* < 0.05; Normal (*n* = 41) versus CRC 41–60 years (*n* = 90), *** *p* < 0.001; Normal (*n* = 41) versus CRC 61–80 years (*n* = 149), **** *p* < 0.0001; Normal (*n* = 41) versus CRC 81–100 years (*n*= 32), ** *p* < 0.01], patients’ weight [Normal (*n* = 41) versus CRC normal weight (*n* = 70), **** *p* < 0.0001; Normal (*n* = 41) versus CRC extreme weight (*n* = 74), **** *p* < 0.0001; Normal (*n* = 41) versus CRC obese (*n* = 56), *** *p* < 0.001; Normal (*n* = 41) versus CRC extreme obese (*n*= 10), ns] was examined by using the UALCAN database.

**Figure 3 cancers-14-05294-f003:**
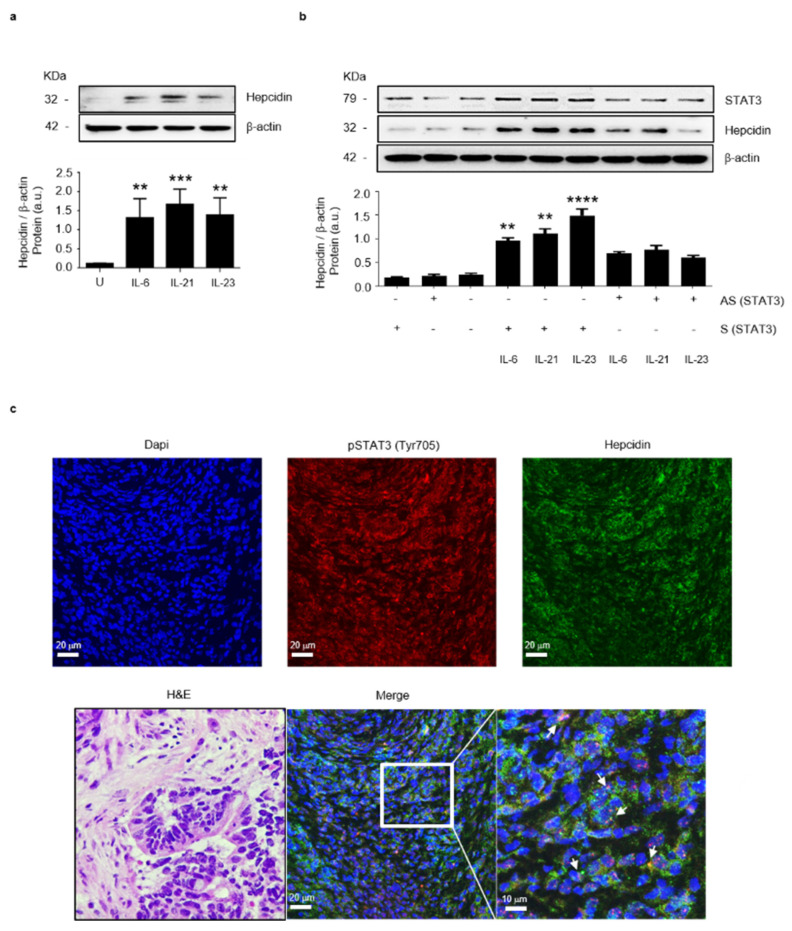
Stat3 regulates positively hepcidin expression in colon epithelial cells. (**a**) Representative Western blots showing hepcidin and β-actin in HCEC-1ct cells stimulated with IL-6, IL-21 and IL-23. Lower panel shows the quantitative analysis of hepcidin/β-actin protein ratio, as measured by densitometry scanning of Western blots of 3 different experiments. Values are expressed in arbitrary units (a.u.) and indicate mean ± SD of all experiments; [Unstimulated (U) versus IL-6, ** *p*< 0.01; U versus IL-21, *** *p* < 0.001; U versus IL-23, ** *p* < 0.01]. (**b**) Representative Western blots showing p-Stat3, hepcidin and β-actin in HCEC-1ct cells transfected with sense (S) or Stat3 antisense (AS) oligonucleotide and then stimulated with IL-6, IL-21 and IL-23, as indicated in materials and methods. Lower panel shows the quantitative analysis of hepcidin/β-actin protein ratio as measured by densitometry scanning of Western blots of 3 different experiments. Values are expressed in arbitrary units (a.u.) and indicate mean ± SD of all experiments; (S/IL-6 versus AS/IL-6, ** *p*< 0.01; S/IL-21 versus AS/IL-21, ** *p* < 0.01; S/IL-23A versus AS/IL-23A, **** *p* < 0.0001; *n* = 3). (**c**) Representative confocal laser scanning microscopy images showing p-Stat3 (Tyr 705) (red) and hepcidin (green) in CRC sections; nuclei are stained with 4′,6-diamidino-2- phenylindole (DAPI) (blue). White arrows indicate cells co-expressing both p-Stat3 and hepcidin. The left image shows the hematoxylin/eosin (H&E) staining of a serial section taken from the same CRC tissue assessed by confocal microscopy.

**Figure 4 cancers-14-05294-f004:**
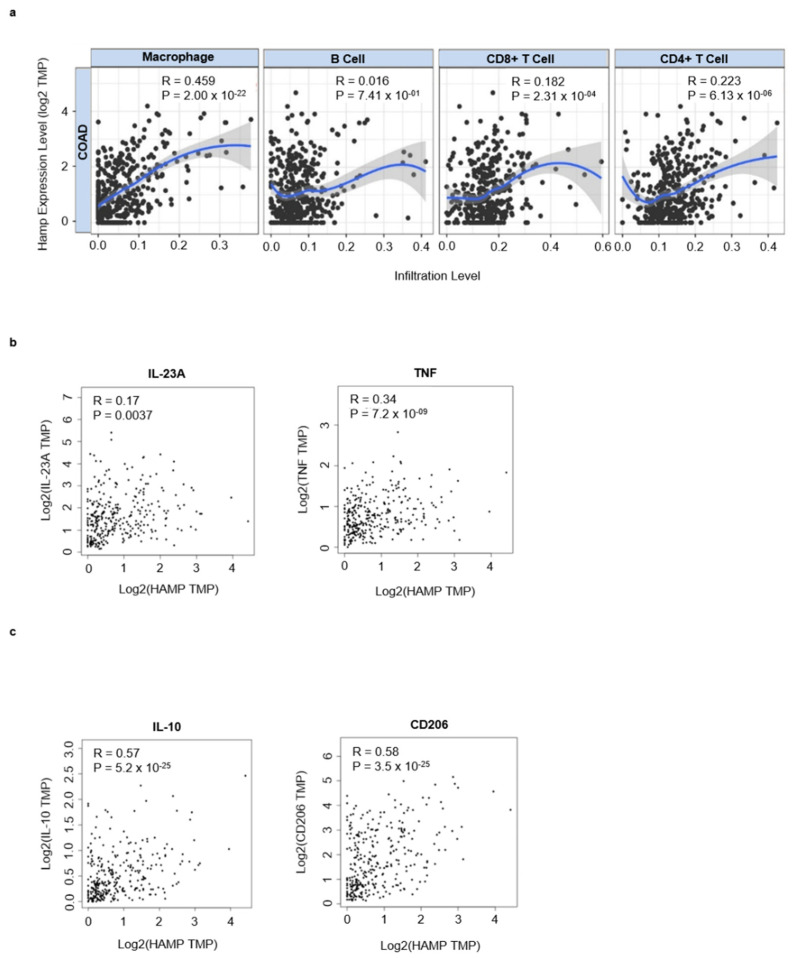
Hepcidin correlates with infiltrating immune cells and macrophages markers in CRC. (**a**) Scatterplots of the correlations between hepcidin expression and the infiltration of different immune cells using the TIMER database. (**b**,**c**) Correlation between hepcidin expression and either IL-23A and TNF, 2 markers of inflammatory macrophages, or IL-10 and CD206, two markers of regulatory macrophages.

**Figure 5 cancers-14-05294-f005:**
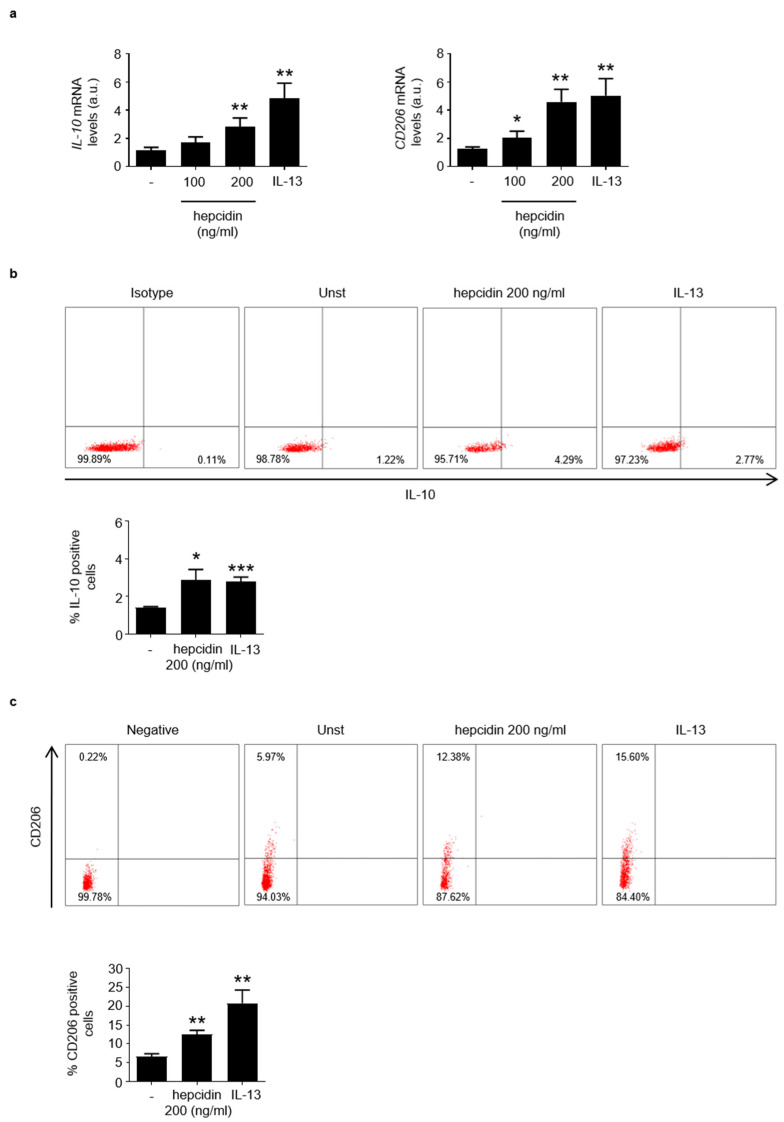
Stimulation of blood mononuclear cells with hepcidin increases the percentages of IL-10- and CD206-expressing cells. (**a**). Blood mononuclear cells were either left unstimulated (Unst) or stimulated with exogenous hepcidin or IL-13 (as an inducer of regulatory macrophages), as indicated in materials. IL-10 and CD206 RNA expression was analyzed by real-time PCR and values were normalized to β-actin RNA. Values are expressed in arbitrary units (a.u.) and indicate mean ± SD of 3 experiments. Unst versus hepcidin, * *p* < 0.05, ** *p* < 0.01; unst versus IL-13, ** *p* < 0.01. (**b**,**c**). Blood mononuclear cells were either left unstimulated (Unst) or stimulated with exogenous hepcidin as indicated in materials and methods and the percentages of IL-10- or CD206-expressing cells were then evaluated by flow-cytometry. The histogram shows the percentage of positive cells and data are expressed as mean ± SEM of 3 separate experiments. Unst versus hepcidin * *p* < 0.05, ** *p* < 0.01; unst versus IL-13, * *p* < 0.0.5, *** *p* < 0.001.

**Figure 6 cancers-14-05294-f006:**
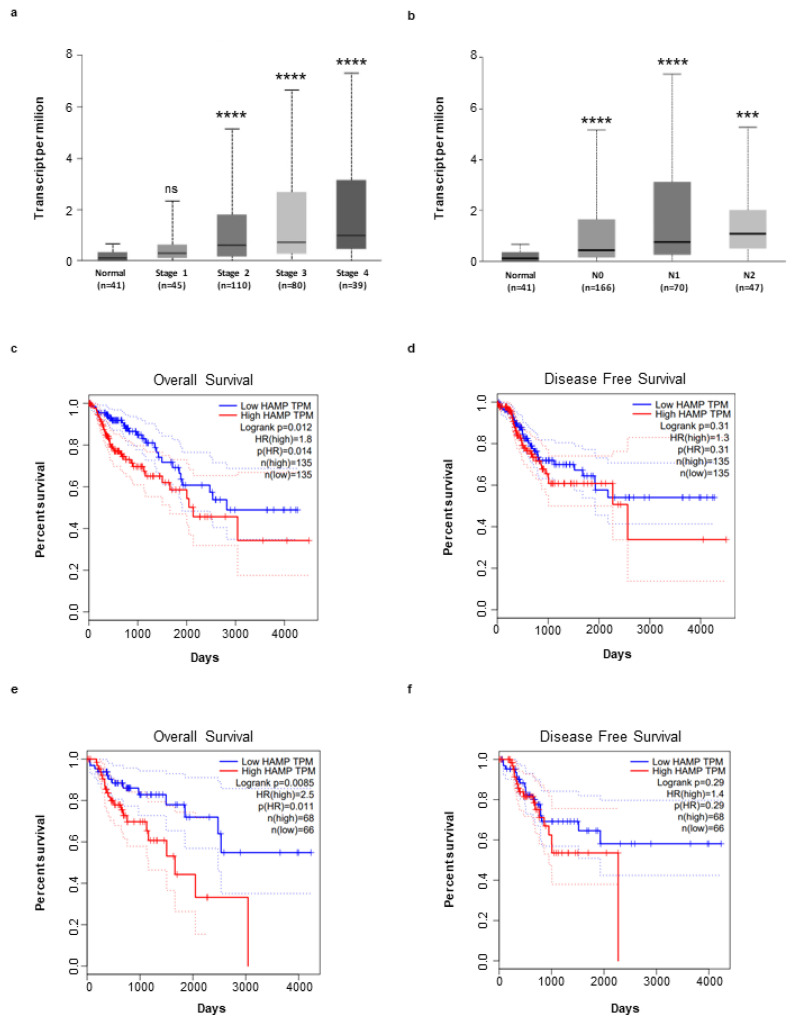
Hepcidin expression is correlated to CRC progression and affects survival of patients. (**a**,**b**) Increase in hepcidin RNA expression in different groups of CRC based on tumor stage (**a**), [Normal (*n* = 41) versus CRC Stage 1 (*n* = 45), ns (ns, not statistical significant); Normal (*n* = 41) versus CRC Stage 2 (*n* = 110), **** *p* < 0.0001; Normal (*n* = 41) versus Stage 3 (*n* = 80), **** *p* < 0.0001; Normal (*n* = 41) versus Stage 4 (*n* = 39), **** *p* < 0.0001] and lymph node metastases (**b**) [Normal (*n* = 41) versus N0 (*n* = 166), **** *p* < 0.0001; Normal (*n* = 41) versus N1 (*n* = 70), **** *p* < 0.0001; Normal (*n* = 41) versus N2 (*n* = 47), *** *p* < 0.001]. Analysis was performed by using the GEIPIA database. (**c**–**f**) Effect of hepcidin expression level on CRC overall survival and disease free survival using the Kaplan–Meier curve (median value: high *n* = 135; low *n* = 135; quartile value: high *n* = 68; low *n* = 68) (broken lines in the graph indicate the lower and upper 95% of confidence intervals) from GEIPIA database.

**Figure 7 cancers-14-05294-f007:**
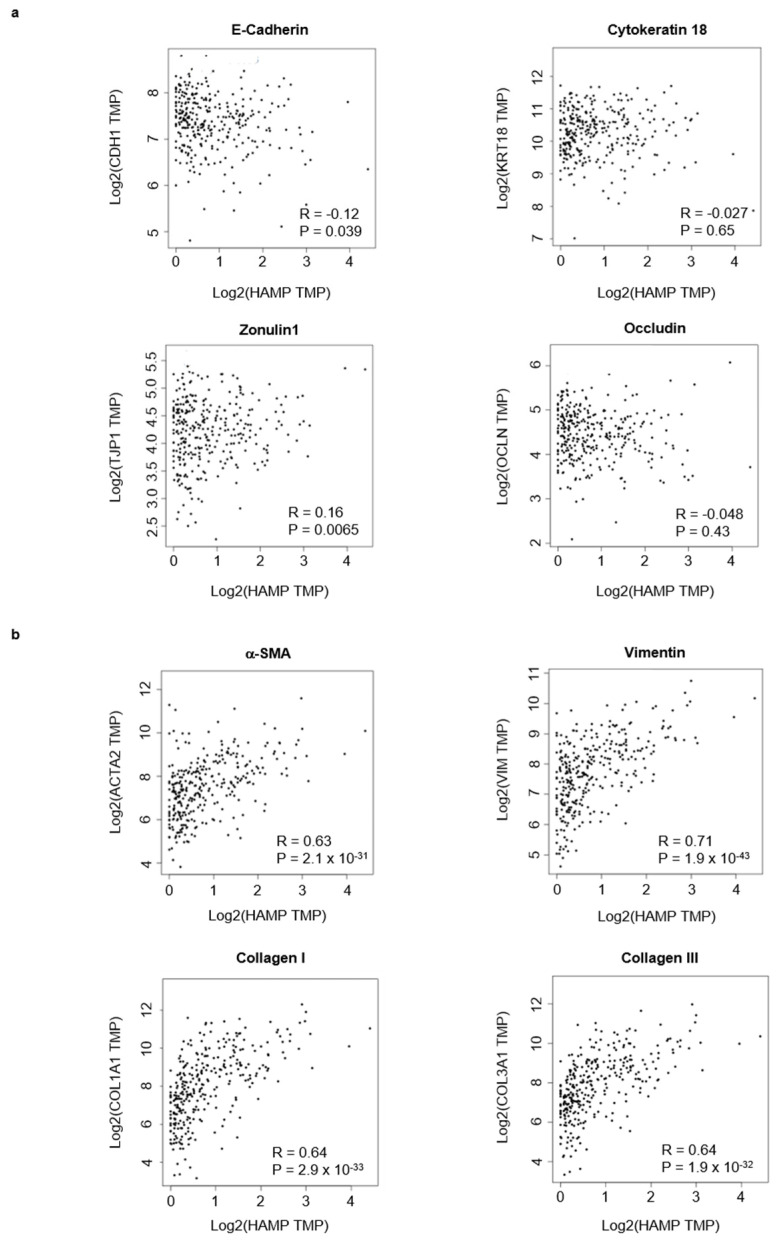
Hepcidin expression correlates with epithelial-to-mesenchymal transition-associated markers in CRC. (**a**) Scatterplots of the correlations between hepcidin expression and epithelial markers using the GEPIA database. (**b**) Scatterplots of the correlations between hepcidin expression and mesenchymal markers using the GEPIA database.

**Figure 8 cancers-14-05294-f008:**
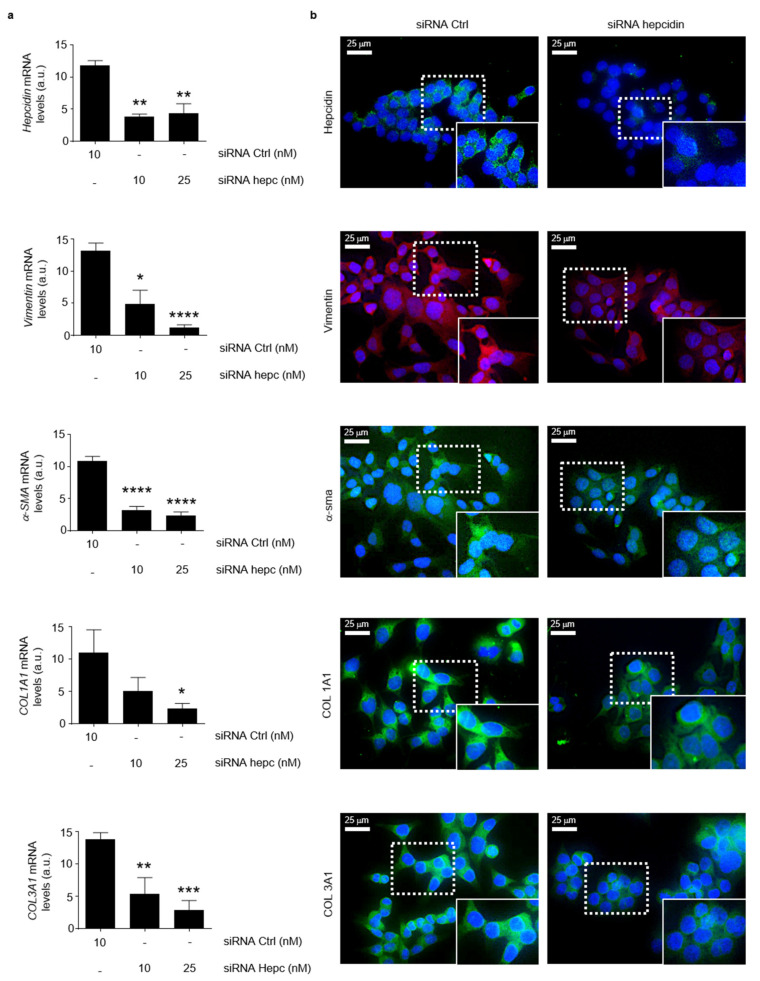
Silencing of hepcidin in CRC cells is associated with a significant reduction of mesenchymal markers. (**a**) Hepcidin, Vimentin, α-SMA, COL1A1 and COL1A3 RNA transcripts were evaluated in HCT-116 cells transfected for 24 h with either a control siRNA (10 nM) or hepcidin siRNA Hepcidin (10 and 25 nM) by real-time PCR, and values were normalized to β-actin RNA. Values are expressed in arbitrary units (a.u.) and indicate mean ± SD of 4 experiments; Hepcidin: control siRNA versus hepcidin siRNA 10 nM, ** *p* < 0.01; control siRNA versus hepcidin siRNA 20 nM, ** *p* < 0.01; Vimentin: control siRNA versus hepcidin siRNA 10 nM, * *p* < 0.05; control siRNA versus hepcidin siRNA 20 nM, **** *p* < 0.0001; α-SMA: control siRNA versus hepcidin siRNA 10 nM, **** *p* < 0.0001; control siRNA versus hepcidin siRNA 20 nM, **** *p* < 0.0001; COL1A1: control siRNA versus hepcidin siRNA 10 nM, ns; control siRNA versus hepcidin siRNA 20 nM, * *p* < 0.05; COL1A3: control siRNA versus hepcidin siRNA 10 nM, ** *p* < 0.01; control siRNA versus hepcidin siRNA 20 nM, *** *p* < 0.001. (**b**) Representative images of immunofluorescence staining for hepcidin (green), Vimentin (red), α-SMA (red), COL1A1 (green) and COL1A3 (green) in HCT-1116 cells transfected with either a control siRNA (10 nM) or hepcidin siRNA Hepcidin (25 nM) for 24 h; nuclei are stained with 4′,6-diamidino-2-phenylindole (DAPI) (blue).

## Data Availability

The datasets supporting the conclusions of this article are included within the article.

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
