# Peer review of "Hepcidin Upregulation in Colorectal Cancer Associates with Accumulation of Regulatory Macrophages and Epithelial–Mesenchymal Transition and Correlates with Progression of the Disease"

_cancers, 2022, doi:10.3390/cancers14215294_

Round 1

Reviewer 1 Report

The paper describes newly identified elevated levels of peptide hormone hepcidin in advanced colorectal cancers. The data derived from UALCAN database appears solid. So is STAT3 and cytokine dependence of hepcidin expression and correlation with macrophage infiltration. Analysis of hepcidin expression by Western blotting in tumor samples and cancer cell lines however raises some questions. The hormone appears on the blots as a 32KDa band, while the molecular weight of the hormone itself is just 2789 Da, and it is only 9408 for the precursor. The discrepancy needs an explanation. There is a likely error in the commercial source of the antibodies to hepcidin as the listed company and its location do not exist.  

Reviewer 2 Report

Hepcidin, a peptide hormone produced mainly by hepatocytes, is expressed by tumor cells and enhances their growth. In the present manuscript, the authors assessed whether hepcidin expression can be used to identify subsets of CRC with advanced and aggressive course. By integrating results of in vitro/ex-vivo studies with data of bioinformatics databases, hepcidin RNA and protein expression was more pronounced in tissue samples taken from the tumor area as compared to the macroscopically unaffected, adjacent, colonic mucosa of CRC patients. Induction of hepcidin in the colonic epithelial cell line HCEC-1ct by interleukin (IL)-6, IL-21 and IL-23 occurred via a Stat3-dependent mechanism and, in primary CRC cells, hepcidin co-localized with active Stat3. In CRC tissues, hepcidin level correlated mainly with macrophage accumulation and IL-10 and CD206 expression, two markers of regulatory macrophages. Consistently, both IL-10 and CD206 were up-regulated by hepcidin in blood mononuclear cells. The highest levels of hepcidin were found in metastatic CRC and survival analysis showed that high expression of hepcidin associated with poor prognosis. Moreover, hepcidin expression correlated with markers of epithelial-to-mesenchymal transition (EMT), and silencing of hepcidin in CRC cells reduced the expression of EMT markers. These findings indicate that hepcidin is markedly induced in the advanced stages of CRC and suggest that it could serve as a prognostic biomarker in CRC.

It is important to identify a new prognostic biomarker for CRC and the authors propose that hepcidin could serve as a prognostic biomarker in CRC. The study lacks detail investigation and some conclusions are not well supported by the results.

Specific comments:

1. Figure 1a: In uninvolved colonic mucosa (NT) of patient 3 had a significant level of hepcidin. Did the authors examine the tissue by IHC? If the authors propose hepcidin could be a biomarker for CRC, it is important to examine cases like this.

2. Figure 1b: In NT of CRC patients, a number of hepcidin-positive cells were seen in the tumor stroma. What are these cells? How many cancer tissues did the authors examine by IHC and how many % of them showed strong staining in cancer cells?

3. Figure 1d: Three CRC cell lines express different levels of hepcidin. What are the features of those cell lines? For example, one may be a more differentiated type than others.

4. Figure 3: When normal colon epithelial cell line cells were stimulated with three cytokines, the level of hepcidin was increased. What is the mechanism whereby hepcidin level is increased in CRC cells? Do they produce hepcidin-inducing cytokines?

5. Figure 3b: The levels of STAT3 and hepcidin in unstimulated control cells and STAT3 KD cells are not presented. What about the effects of STAT3 KD on cell survival and proliferation?

6. Figure 3c: Are the cells positive for hepcidin and STAT3 cancer cells or stromal cells? It’s not clear from the photos.

7. Page 9: “There was a positive correlation between hepcidin RNA expression and markers associated with either inflammatory (i.e. TNF and IL-23A) or regulatory macrophages (i.e. IL-10 and CD206).” Did the authors compare mRNA levels? Not cell number?

8. Figure 5: Only a small percentage of the cells responded to hepcidin and became positive for IL-10 or CD206. Why? Controls that induce M1 or M2 polarization should be used to evaluate the activity of hepcidin to induce M2 polarization. Effects on M1 polarization were not examined.

9. Figure 6a: Which stages include LN involvement? Analysis using ANOVA may be necessary to compare among stages.

10. Figure 6b: “Significantly higher hepcidin RNA levels were seen in CRC patients compared to controls independently of the lymph node involvement (Figure 6b). “These results suggest that hepcidin expression marks a subset of patients with metastatic disease.” These two sentences seem not consistent.

11. Dose hepcidin induce STAT3 activation in CRC cells?

12. Figure 6c-e: What are broken lines?

13. Figure 8: “silencing of hepcidin in HCT-116 cells led to a significant reduction of the mesenchymal markers compared with control samples.” What about the effects on cell shape or other functions of cells with EMT? 

14. Page 14, line 5-7 from the bottom: “These observations raise the possibility that different cytokines released by immune cells infiltrating the CRC tissue activate Stat3 in cancer cells, thereby promoting hepcidin synthesis.” Silencing hepcidin reduced the expression of EMT markers in HCT-116 cells. What is the mechanism by which the expression of hepcidin is upregulated in this cell line? Do they produce one of STAT3-activating cytokines?

15. Labels are too small. Use a larger font size.

Reviewer 3 Report

The authors demonstrated that Hepcidin upregulation in colorectal cancer associates with accumulation of regulatory macrophages, epithelial-mesen-chymal transition, and progression of the cancer. The authors employed a lot of analyses by using online database, and thus, the experimental data is not sufficient to verify these analyses. There are some issues need to be addressed.

1. The author found the colocalization between p-STAT3 and hepcidin; however, the author should use confocal microscope to detect the colocalization. Moreover, the author should expain the significance or mechanism about the colocalization or binding of these proteins.

2. The author use lots of R value to analyze the gene expression correlation, the author should provide the standard to conclude positive or nagative correlation, as some R value are even lower than 0.2. 

3. The author should provide the detail of the method to analyze the percentage of IL-10 positive cells in Figure 5, as FACS is not considered the proper method to detect the intracelluar protein.

4.  The author found both inflammatory and regulatory macrophages, which may referred to  M1 and M2 macrophage, the author should provide more data by using qPCR and immunoflourescence.  

5. Western blotting should be performed besides qPCR in Figure 8.

Round 2

Reviewer 2 Report

Response to comment 8:

It should be noted in the manuscript that the authors examined the expression of M1 markers and did not see differences.

Response to comment 13:

Shape change is a feature of EMT. For example, from cobble stone shape to elongated spindle shape. In this case, from elongated spindle shape to cobble stone shape after siRNA treatment. This reviewer was asking whether the authors observed a simple shape change. It is not clear from the figure. If there was no change, the authors should state so.

Author Response

We would like to thank the reviewer for his/her suggestions.

In response to the specific questions of this reviewer:

Response to comment 8:

 It should be noted in the manuscript that the authors examined the expression of M1 markers and did not see differences.

Response: We have changed the manuscript and specified that treatment of PBMC with exogenous hepcidin did not change TNF RNA expression.

Response to comment 13:

Shape change is a feature of EMT. For example, from cobble stone shape to elongated spindle shape. In this case, from elongated spindle shape to cobble stone shape after siRNA treatment. This reviewer was asking whether the authors observed a simple shape change. It is not clear from the figure. If there was no change, the authors should state so.

Response: We do agree with the reviewer. To avoid any confusion we have revised the text and made clear that, at the specified time point, hepcidin silencing associated with no significant change in HCT116 cell shape. 

Reviewer 3 Report

The author did not adequately answer all the questions, and the manuscript may not be accepted in its current form. The questions below still need to be addressed.

  1.    STAT3 as a transcription factor can regulate the expression of hepcidin; however, the colocalization of these proteins did not provide any clue for the regulation. Also, the immunofluorescence was not solid data for the colocalization of these proteins. Therefore, the above co-localization has no relationship with the underlying mechanism of Hepcidin upregulation, the author should better remove these data.   2.    The R values less than 0.2 are too low to be accepted, if the author thinks R-value lower than 0.2 is acceptable for a positive association, please provide some reference.

    3. The author still did not provide the detail of the method to analyze the percentage of IL-10 positive cells in Figure 5, as FACS usually detect the protein in the cell membrane but not intercellular proteins.

Author Response

We would like to thank the reviewer for his/her evaluation. In response to the specific questions of this reviewer:

The author did not adequately answer all the questions, and the manuscript may not be accepted in its current form. The questions below still need to be addressed.

  1. STAT3 as a transcription factor can regulate the expression of hepcidin; however, the colocalization of these proteins did not provide any clue for the regulation. Also, the immunofluorescence was not solid data for the colocalization of these proteins. Therefore, the above co-localization has no relationship with the underlying mechanism of Hepcidin upregulation, the author should better remove these data.  

Response. We agree with reviewer that co-localisation of active Stat3 and hepcidin is not sufficient to prove that Stat3 signaling controls hepcidin expression. In fact, we performed a lot of studies to prove that activation of Stat3 with cytokines enhances hepcidin induction as well as inhibition of Stat3 down-regulates hepcidin expression (see figure 3). 

  1. The R values less than 0.2 are too low to be accepted, if the author thinks R-value lower than 0.2 is acceptable for a positive association, please provide some reference.

Response. Although it is weak,  the R value of 0.2 indicates always a positive association. We do not think it is relevant to provide references to support the validity of a statistical test.  

  1. The author still did not provide the detail of the method to analyze the percentage of IL-10 positive cells in Figure 5, as FACS usually detect the protein in the cell membrane but not intercellular proteins.

Response: This is not true as details of Flow-cytometry assay were reported in the previous and current version of the manuscript (see paragraph 2.6  Flow cytometry).

We do not agree with reviewer that FACS detects only cell membrane-associated proteins. The literature is full of papers showing the ability of this procedure to assess intracellular proteins. Indeed, FACS is typically used to assess the expression of transcription factors. Probably, the reviewer is confusing what we stated in the manuscript, because we used FACS to assess the intracellular content of IL-10 and not the intercellular level of the protein.